analytical chemistry/plant science

dandelion, quality evaluation, quantitative analysis of multi-components by single marker, HPLC

**Authors for correspondence:**
Chunjian Zhao
e-mail: zcj@nefu.edu.cn
Yujie Fu
e-mail: yujie_fu@163.com

This article has been edited by the Royal Society of Chemistry, including the commissioning, peer review process and editorial aspects up to the point of acceptance.

# Application of fingerprint combined with quantitative analysis and multivariate chemometric methods in quality evaluation of dandelion (*Taraxacum mongolicum*)

Chunying Li[1,2], Yao Tian[1,2], Chunjian Zhao[1,2], Shen Li[1,2], Tingting Wang[1,2], Bin Qiao[1] and Yujie Fu[1,2]

[1]Key Laboratory of Forest Plant Ecology, Ministry of Education, and [2]College of Chemistry, Chemical Engineering and Resource Utilization, Northeast Forestry University, Harbin 150040, People's Republic of China

CL, 0000-0003-1583-6436; CZ, 0000-0003-2827-1999

A quality assessment method based on quantitative analysis of multi-components by single marker (QAMS) and fingerprint was constructed from 15 batches of dandelion (*Taraxacum mongolicum*), using multivariate chemometric methods (MCM). MCM were established by hierarchical cluster analysis (HCA) and factor analysis (FA). HCA was especially performed using the R language and SPSS 22.0 software. The relative correction factors of chlorogenic acid, caffeic acid, p-coumaric acid, luteolin and apigenin were calculated with cichoric acid as a reference, and their contents were determined. The differences between external standard method (ESM) and QAMS were compared. There was no significant difference (*t*-test, $p > 0.05$) in quantitative determination, proving the consistency of the two methods (QAMS and ESM). Dandelion material from Yuncheng, Shandong was used as a reference chromatogram. The fingerprints in 15 batches of dandelion were established by HPLC analysis. The similarity of the fingerprints in different batches of dandelion material was greater than or equal to 0.82. A total of 10 common peaks were identified. This strategy is simple, rapid and efficient in multiple component detection of dandelion. It is beneficial in simplifying dandelion's quality control processes and providing references to enhance quality control for other herbal medicines.

# 1. Introduction

Dandelion (*Taraxacum mongolicum* Hand.-Mazz.) is a perennial plant in the Composite family. Its flowering period is from April to October [1]. Dandelion was distributed widely in many countries. There are more than 2000 kinds of varieties of dandelion; about 70 kinds of varieties are distributed in various provinces in China [2]. The edible value, medical value and nutritional value of dandelion have been highly appraised and affirmed in Compendium of Materia Medica and other ancient medical ceremonies [3]. The edible portion of dandelion reaches 84%; the leaves of dandelion, consumed as vegetable food, contains vitamin C, vitamin D, carotene and a lot of iron, calcium and other nutrients [4]. Dandelion has been reported to slow down the damage by the effects of oxygen [5], suppress or reduce inflammation [6], fight against cancer [7], resist high concentration of sugar in the blood [8], prevent or impair coagulation [9], soothe soreness [10] and reduce the pathological reaction caused by strong stimulation of the body [11].

Dandelion is rich in phenolic compounds and flavonoids compounds, which are known to promote health [12]. At present, HPLC and HPLC-MS have been used for qualitative and quantitative analysis of the main bioactive components of dandelion [13,14]. These external standard methods (ESM) rely on relative retention time, weak ultraviolet absorption, complex background interference and other shortcomings, which limit the application of these methods [15–17]. Above all, ESM was unable to concurrently determine multiple components in the target sample, resulting in a complicated process and low efficiency [18]. The quantitative analysis of multi-components by single marker (QAMS) only needs to select a reference in the sample. Establishing its relationship with other components in the sample can make the simultaneous determination of the content of multiple components become feasible [19]. This could reduce the time and cost spent in the quality control of herbaceous plant products and bring about ulteriorly improving the HPLC practicability [20,21]. Therefore, QAMS has extensive adhibition to regulate the quality of traditional Chinese medicine (TCM) [22], but it has not been reported in quality control of dandelion.

Recently, researchers used the chromatographic fingerprint to analyse the quality of TCM; it has been approved by many national drug administrations (FDA, SFDA, EMA) [23]. Chromatographic fingerprint method was used to identify substitutes and adulterants according to a limited number of characteristic peaks of genuine materials [24], but the characteristic fingerprint cannot give expression to the content of the active natural ingredients of dandelion. The whole information of dandelion is blurred by the characteristic fingerprint, and the multi-components of dandelion need to be determined. The combination of characteristic fingerprint and QAMS by multivariate chemometric methods (MCM) was used to compare the similarity of dandelion fingerprint. MCM were established by hierarchical cluster analysis (HCA) and factor analysis (FA) [25], and HCA was especially performed using the R language and SPSS 22.0 software.

# 2. Material and methods

## 2.1. Chemicals and materials

A total of dandelion samples (S1–S15) were collected from different Chinese provinces. Table 1 lists the detailed local information. Six standard controls (chlorogenic acid, caffeic acid, p-coumaric acid, cichoric acid, luteolin and apigenin) with purity greater than 98% were from Chengdu MUST Biotech Co., Ltd, [26]. The HPLC grade formic acid, acetonitrile and methanol were acquired from DIKMA Technologies (Beijing, China). Other chemicals used in the experiment were from Tianjin Tianli Reagents Co., Ltd (Tianjin, China).

## 2.2. Instruments and chromatographic conditions

The analytical instrument was Agilent 1260 series HPLC device. Analytes were separated by Eco-silC18 column (5 μm, 250 × 4.6 mm). The HPLC system stood a flow rate of 0.8 ml min$^{-1}$; the column temperature was settled as 35°C and the injection volume of the sample was set as 10 μl. The measurement wavelength was set at 254 nm. Mobile phase A was 0.2% phosphoric acid aqueous solution and B was acetonitrile. The elution gradient was 0–5 min, 20–27% B; 5–12 min, 27–32% B; 12–14 min, 32–34% B; 14–17 min, 34–37% B; 17–27 min, 37–45% B.

**Table 1.** The different geographical locations, similarities, score and comprehensive evaluation results of 15 batches of dandelion in China.

| no. | district | similarity | score | ranking |
|---|---|---|---|---|
| S1 | Baishan City, Jilin Province | 0.958 | 2.408 | 3 |
| S2 | Baishan City, Jilin Province | 0.982 | 0.956 | 13 |
| S3 | Bozhou City, Anhui Province | 0.952 | 1.519 | 9 |
| S4 | Bozhou City, Anhui Province | 0.951 | 2.500 | 2 |
| S5 | Changbai Mountain City, Jilin Province | 0.882 | 0.551 | 15 |
| S6 | Chengdu City, Sichuan Province | 0.919 | 1.797 | 6 |
| S7 | Hulun Buir City, Inner Mongolia Autonomous Region | 0.989 | 1.600 | 7 |
| S8 | Jinan City, Shandong Province | 0.963 | 1.533 | 8 |
| S9 | Kunming City, Yunnan Province | 0.975 | 1.011 | 11 |
| S10 | Laiyang City, Shandong Province | 0.832 | 0.971 | 12 |
| S11 | Laiyang City, Shandong Province | 0.829 | 0.676 | 14 |
| S12 | Lanxi County, Heilongjiang Province | 0.952 | 1.368 | 10 |
| S13 | Linyi City, Shandong Province | 0.961 | 2.223 | 4 |
| S14 | Nanjing City, Jiangsu Province | 0.984 | 2.647 | 1 |
| S15 | Yuncheng City, Shandong Province | 0.964 | 2.071 | 5 |

## 2.3. Preparation of sample solutions

One gram of dandelion powders was accurately weighed. It was soaked into 30 ml of 70% methanol–water solution, placed in a conical flask and ultrasonication (25°C, 250 W, 60 kHz) performed for 30 min. After the extract was fully mixed and shaken, the centrifugation was carried out at a fast speed of 10 000 r.p.m. The collected supernatant was filtered with a 0.45 µm filter membrane, and the obtained sample solution could be directly analysed by HPLC.

## 2.4. Preparation of standard solution

The chlorogenic acid, caffeic acid, p-coumaric acid, cichoric acid and luteolin standard references were weighed and dissolved into a standard solution of 1.0 mg ml$^{-1}$ with methanol. The apigenin standard reference was weighed and dissolved into 0.5 mg ml$^{-1}$ solution with methanol. The mixed standard solution was procured by blending 0.2 ml of the individual stock solutions. Except the concentration of apigenin was 0.083 mg ml$^{-1}$, the other standard reference concentrations were 0.167 mg ml$^{-1}$.

## 2.5. Computation of relative conversion factors

There are a variety of components in the sample. Among these components, which being stable, easy to obtain and separate from other components are selected as a single marker, so that a single marker can accurately determine other multiple components. And simultaneously cichoric acid is rich in dandelion [27], thus, it is suitable for the quality indicator of dandelion. Using cichoric acid as a single marker [28], the factor ratio of a single factor marker with other analytes is $f_{si}$ using formula (2.1) [29]. The concentration of each other analyte ($C_i$) in the sample could be calculated according to formula (2.2) [30],

$$f_{si} = \frac{f_s}{f_i} = \frac{A_s/C_s}{A_i/C_i} \tag{2.1}$$

and

$$C_i = f_{si} \times C_s \times \frac{A_i}{A_s}. \tag{2.2}$$

$A_s$ is the peak area of cichoric acid and $A_i$ is the peak area of other analytes. $C_s$ is the concentration of cichoric acid and $C_i$ is the concentration of other analytes (mg ml$^{-1}$).

**Table 2.** Horizontal table of orthogonal test factors.

| no. | solid–liquid ratio | concentration of solvent | extracting time | column temperature | total |
|---|---|---|---|---|---|
| 1 | 1 : 25 | 60% | 15 | 30 | 17.013 |
| 2 | 1 : 25 | 70% | 30 | 35 | 18.898 |
| 3 | 1 : 25 | 80% | 45 | 40 | 15.245 |
| 4 | 1 : 30 | 60% | 30 | 40 | 19.402 |
| 5 | 1 : 30 | 70% | 45 | 30 | 19.983 |
| 6 | 1 : 30 | 80% | 15 | 35 | 16.974 |
| 7 | 1 : 35 | 60% | 45 | 35 | 18.930 |
| 8 | 1 : 35 | 70% | 15 | 40 | 19.912 |
| 9 | 1 : 35 | 80% | 30 | 30 | 16.176 |
| $k1$ | 17.052 | 18.448 | 17.966 | 17.724 | |
| $k2$ | 18.786 | 19.597 | 18.159 | 18.267 | |
| $k3$ | 18.339 | 16.132 | 18.053 | 18.186 | |
| $R$ | 1.734 | 3.465 | 0.193 | 0.543 | |

$Ki$ = the sum of the index values of the numbers in column 5 and 'i'.
$R$ = The difference between the maximum and the minimum of the average values of $K1$, $K2$… in column 5.

## 2.6. Statistical analysis

The data were analysed and evaluated by a similarity evaluation system for the chromatographic fingerprint of TCM (2012, China), which was recommended by SFDA [31]. The similarity among different chromatograms was quantified by calculating the correlative coefficient. The similarity between the samples was acquired by computing the correlation coefficients of different chromatograms. R language conducts HCA according to the similarity degree of each component among different samples. IBM SPSS Statistical 22.0 software (IBM, New York, USA) applies the square Euclidean distance computing of the content of each component in the sample to perform HCA. HCA based on R language and SPSS distinguish herbal species. In order to verify the feasibility of QAMS, the other five active components in dandelion samples were determined by applying cichoric acid as an internal reference.

# 3. Results and discussion

## 3.1. Screening of chromatographic conditions

The suitable extraction method and HPLC parameters were tested, and the optimal chromatographic fingerprint was finally obtained. We got the optimized extraction efficiency by three column temperatures (30°C, 35°C, 40°C), solid–liquid ratio (1 : 25, 1 : 30, 1 : 35 g ml$^{-1}$), concentration of solvent (60%, 70%, 80%), extracting time (15, 30, 45 min). One gram of dandelion powder was soaked in 70% methanol–water ultrasonication for 30 min. It was simpler and more effective for the extraction of dandelion (table 2). Finally, the gradient solvent system consisted of 0.2% phosphoric acid in water (eluent A) and acetonitrile (eluent B) was at a column temperature of 35°C with a flow rate of 0.8 ml min$^{-1}$; the detection wavelength was set at 254 nm. The above conditions were given the necessary best performance (reconstruction and separation) in a chromatographic fingerprint.

## 3.2. Method validation

### 3.2.1. Linearity

Six standard solutions (chlorogenic acid, caffeic acid, p-coumaric acid, cichoric acid, luteolin and apigenin) were diluted with methanol to six different concentrations. According to the relationship between the peak area (Y) and the concentration of each analyte (X), the partial least square method

**Table 3.** The regression data and linear range for six bioactive compounds analysed by HPLC ($n = 6$).

| standard solutions | regression equations | $R^2$ | linear range ($\mu$g ml$^{-1}$) |
|---|---|---|---|
| chlorogenic acid | $y = 10.69x + 48.25$ | $R^2 = 0.9991$ | 7.5–100.0 |
| caffeic acid | $y = 35.03x - 56.98$ | $R^2 = 0.9991$ | 2.5–163.0 |
| P-coumaric acid | $y = 21.53x - 76.26$ | $R^2 = 0.9993$ | 1.5–200.0 |
| cichoric acid | $y = 29.75x + 23.08$ | $R^2 = 0.9990$ | 75.0–525.0 |
| luteolin | $y = 91.02x - 28.30$ | $R^2 = 0.9997$ | 2.0–175.0 |
| apigenin | $y = 22.08x + 49.15$ | $R^2 = 0.9996$ | 1.6–87.50 |

**Table 4.** RSD of precision, stability, repeatability and accuracy for determination of six components ($n = 6$).

| standard solutions | precision RSD (%) | stability RSD (%) | repeatability RSD (%) | accuracy mean (%) | accuracy RSD (%) |
|---|---|---|---|---|---|
| chlorogenic acid | 1.76 | 1.23 | 2.05 | 99.82 | 1.15 |
| caffeic acid | 1.66 | 1.40 | 0.47 | 100.25 | 2.04 |
| P-coumaric acid | 1.08 | 1.35 | 2.11 | 99.89 | 1.51 |
| cichoric acid | 1.30 | 2.45 | 1.66 | 101.24 | 2.37 |
| luteolin | 2.03 | 2.21 | 1.24 | 103.33 | 2.10 |
| apigenin | 0.89 | 1.32 | 3.42 | 100.10 | 2.25 |

**Table 5.** The results of RCF ($f_{si}$).

| flow rate | column temperature | chlorogenic acid | caffeic acid | p-coumaric acid | luteolin | apigenin |
|---|---|---|---|---|---|---|
| 0.6 | 30 | 2.7026 | 0.8157 | 1.2796 | 0.3808 | 0.5750 |
| 0.6 | 35 | 2.7050 | 0.8162 | 1.2833 | 0.3862 | 0.5675 |
| 0.6 | 40 | 2.7036 | 0.8135 | 1.2842 | 0.3851 | 0.5759 |
| 0.8 | 30 | 2.7112 | 0.8198 | 1.2787 | 0.3823 | 0.5746 |
| 0.8 | 35 | 2.7033 | 0.8173 | 1.2820 | 0.3811 | 0.5802 |
| 0.8 | 40 | 2.6997 | 0.8187 | 1.2829 | 0.3833 | 0.5776 |
| 1 | 30 | 2.7028 | 0.8190 | 1.2814 | 0.3824 | 0.5754 |
| 1 | 35 | 2.7031 | 0.8211 | 1.2835 | 0.3835 | 0.5721 |
| 1 | 40 | 2.6981 | 0.8179 | 1.2830 | 0.3836 | 0.5760 |
| means | | 2.7033 | 0.8177 | 1.2821 | 0.3831 | 0.5749 |
| RSD% | | 0.13 | 0.27 | 0.14 | 0.43 | 0.58 |

was used to draw the linear regression equation ($Y = aX + b$). The linear regression equation could be applied to QAMS analysis (table 3).

### 3.2.2. Precision, stability, repeatability and accuracy

The precision was assessed by analytic judgement of the same solution of six standards ($n = 6$) within one day. The results showed that the relative standard deviations (RSDs) of chlorogenic acid, caffeic acid, p-coumaric acid, cichoric acid, luteolin and apigenin were 1.76%, 1.66%, 1.08%, 1.30%, 2.03% and 0.89% ($n = 6$), respectively. It indicated that the precision of the method was good.

**Table 6.** The determination of six components in 15 batches of dandelion between QAMS and ESM (mg g$^{-1}$). RE represents relative error. RSD represents relative standard deviation. $p$-values represent the paired $t$-test results.

| no. | dichoric acid | chlorogenic acid | | | | caffeic acid | | | | p-coumaric acid | | | | luteolin | | | | apigenin | | | |
|---|---|---|---|---|---|---|---|---|---|---|---|---|---|---|---|---|---|---|---|---|---|
| | ESM | ESM | QAMS | RE% | RSD% | ESM | QAMS | RE% | RSD% | ESM | QAMS | RE% | RSD% | ESM | QAMS | RE% | RSD% | ESM | QAMS | RE% | RSD% |
| S1 | 14.148 | 1.203 | 1.209 | 0.50 | 0.35 | 0.250 | 0.249 | −0.40 | 0.28 | 0.249 | 0.249 | 0.00 | 0.21 | 0.003 | 0.003 | 0.00 | 0.33 | 0.015 | 0.015 | 0.00 | 0.77 |
| S2 | 5.136 | 0.678 | 0.654 | −3.54 | 2.55 | 0.147 | 0.150 | 2.04 | 1.43 | 0.282 | 0.285 | 1.06 | 0.75 | 0.006 | 0.006 | 0.00 | 0.98 | 0.003 | 0.003 | 0.00 | 0.43 |
| S3 | 9.177 | 0.600 | 0.603 | 0.50 | 0.35 | 0.329 | 0.324 | −1.52 | 1.08 | 0.075 | 0.078 | 4.00 | 2.77 | 0.009 | 0.009 | 0.00 | 0.40 | 0.006 | 0.006 | 0.00 | 0.15 |
| S4 | 14.949 | 0.963 | 0.948 | −1.56 | 1.11 | 0.423 | 0.426 | 0.71 | 0.50 | 0.276 | 0.276 | 0.00 | 0.59 | 0.015 | 0.015 | 0.00 | 0.65 | 0.003 | 0.003 | 0.00 | 0.56 |
| S5 | 2.856 | 0.225 | 0.228 | 1.33 | 0.94 | 0.621 | 0.624 | 0.48 | 0.34 | 0.060 | 0.06 | 0.00 | 0.23 | 0.162 | 0.162 | 0.00 | 0.71 | 0.093 | 0.090 | −3.23 | 2.32 |
| S6 | 11.298 | 0.327 | 0.342 | 4.59 | 3.17 | 0.369 | 0.375 | 1.63 | 1.14 | 0.054 | 0.054 | 0.00 | 0.71 | 0.012 | 0.012 | 0.00 | 0.67 | 0.006 | 0.006 | 0.00 | 0.49 |
| S7 | 9.114 | 0.717 | 0.702 | −2.09 | 1.49 | 0.388 | 0.381 | −1.80 | 1.29 | 0.309 | 0.312 | 0.97 | 0.68 | 0.093 | 0.090 | −3.23 | 2.32 | 0.018 | 0.018 | 0.00 | 0.29 |
| S8 | 8.643 | 0.945 | 0.936 | −0.95 | 0.68 | 0.126 | 0.129 | 2.38 | 1.66 | 0.276 | 0.282 | 2.17 | 1.52 | 0.003 | 0.003 | 0.00 | 1.32 | 0.012 | 0.012 | 0.00 | 0.78 |
| S9 | 5.508 | 0.648 | 0.636 | −1.85 | 1.32 | 0.122 | 0.120 | −1.64 | 1.17 | 0.288 | 0.291 | 1.04 | 0.73 | 0.009 | 0.009 | 0.00 | 0.46 | 0.003 | 0.003 | 0.00 | 0.45 |
| S10 | 5.199 | 0.498 | 0.498 | 0.00 | 0.10 | 0.699 | 0.711 | 1.72 | 1.20 | 0.201 | 0.198 | −1.49 | 1.06 | 0.102 | 0.099 | −2.94 | 2.11 | 0.087 | 0.090 | 3.45 | 2.40 |
| S11 | 3.135 | 0.657 | 0.642 | −2.28 | 1.63 | 0.540 | 0.531 | −1.67 | 1.19 | 0.222 | 0.219 | −1.35 | 0.96 | 0.075 | 0.078 | 4.00 | 2.77 | 0.093 | 0.090 | −3.23 | 2.32 |
| S12 | 7.890 | 0.789 | 0.759 | −3.80 | 2.74 | 0.784 | 0.792 | 1.02 | 0.72 | 0.117 | 0.114 | −2.56 | 1.84 | 0.015 | 0.015 | 0.00 | 0.74 | 0.003 | 0.003 | 0.00 | 0.38 |
| S13 | 12.723 | 1.332 | 1.296 | −2.70 | 1.94 | 0.309 | 0.303 | −1.94 | 1.39 | 0.285 | 0.279 | −2.11 | 1.50 | 0.057 | 0.057 | 0.00 | 0.58 | 0.006 | 0.006 | 0.00 | 1.42 |
| S14 | 15.453 | 0.966 | 0.975 | 0.93 | 0.66 | 0.352 | 0.345 | −1.99 | 1.42 | 0.324 | 0.318 | −1.85 | 1.32 | 0.12 | 0.117 | −2.5 | 1.79 | 0.177 | 0.180 | 1.69 | 1.19 |
| S15 | 11.688 | 0.738 | 0.756 | 2.44 | 1.70 | 0.614 | 0.612 | −0.33 | 0.23 | 0.354 | 0.348 | −1.69 | 1.21 | 0.171 | 0.174 | −1.69 | 1.23 | 0.213 | 0.210 | −1.41 | 1.00 |
| max | 15.453 | 1.332 | 1.296 | | | 0.784 | 0.792 | | | 0.354 | 0.348 | | | 0.171 | 0.174 | | | 0.213 | 0.210 | | |
| min | 2.856 | 0.222 | 0.228 | | | 0.122 | 0.120 | | | 0.054 | 0.054 | | | 0.003 | 0.003 | | | 0.003 | 0.003 | | |
| means | 9.1278 | 0.7524 | 0.7456 | | | 0.4049 | 0.4048 | | | 0.2248 | 0.2242 | | | 0.0568 | 0.0566 | | | 0.0492 | 0.049 | | |
| correlation coefficient | | | 0.999** | | | | 0.999** | | | | 0.999** | | | | 0.999** | | | | 0.999** | | |
| $p$-values | | | 0.126 | | | | 0.967 | | | | 0.550 | | | | 0.670 | | | | 0.678 | | |

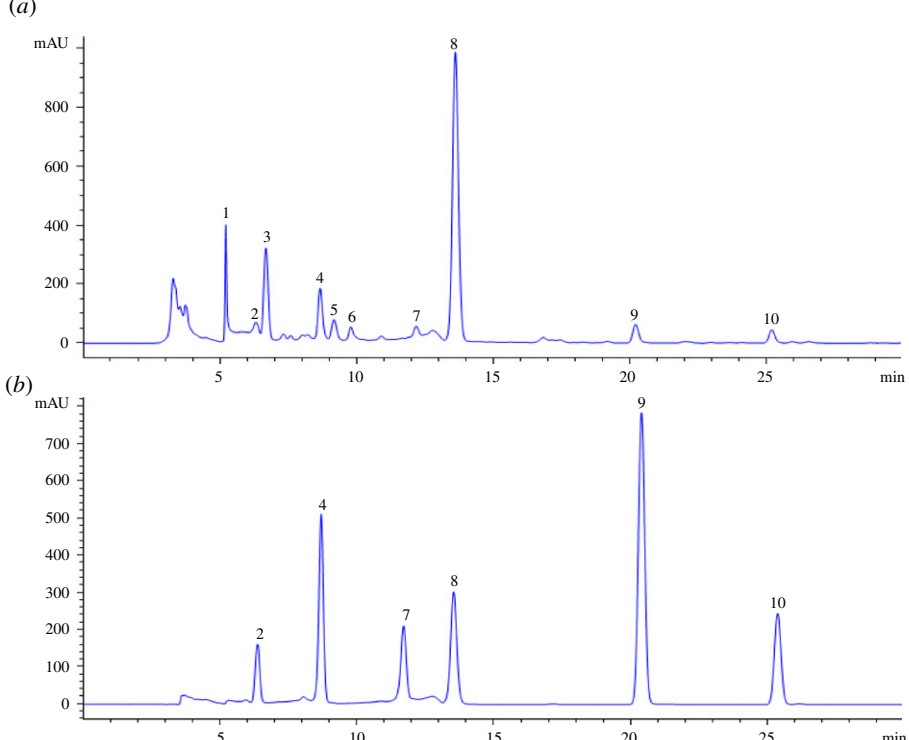

**Figure 1.** HPLC fingerprint of dandelion and mixed standard. (*a*) the dandelion sample, (*b*) the mixed standards. 2: chlorogenic acid, 4: caffeic acid, 7: p-coumaric acid, 8: cichoric acid, 9: luteolin, 10: apigenin.

The stabilities of the same sample solutions (S15) were analysed at 0, 2, 6, 8, 16 and 24 h after storage for one day (25°C). The RSD values for the stability tests of chlorogenic acid, caffeic acid, p-coumaric acid, cichoric acid, luteolin and apigenin were 1.23%, 1.40%, 1.35%, 2.45%, 2.21% and 1.32% ($n = 6$), respectively. It suggested that the method was steady within 24 h.

Take the same batch of samples (S15), according to the method of sample preparation, each inject 10 µl, respectively ($n = 6$). The results showed that the RSDs of chlorogenic acid, caffeic acid, p-coumaric acid, cichoric acid, luteolin and apigenin were 2.05%, 0.47%, 2.11%, 1.66%, 1.24% and 3.42%, ($n = 6$), respectively. It indicated that the reproducibility of the method was good.

Low, medium and high concentrations of the mixed standard were added into dandelion samples (S15), to determine the accuracy of the method. The average recovery of chlorogenic acid, caffeic acid, p-coumaric acid, cichoric acid, luteolin and apigenin were 99.82%, 100.25%, 99.89%, 101.24%, 103.33% and 100.10%, and the RSDs were 1.15%, 2.04%, 1.51%, 2.37%, 2.10% and 2.25%, respectively. It demonstrated the method was accurate (table 4).

## 3.3. The evaluation of quantitative analysis of multi-components by single marker and external standard method

In order to assess and validate QAMS feasibility for the determination of multi-compounds in dandelion, the contents of chlorogenic acid, caffeic acid, p-coumaric acid, cichoric acid, luteolin and apigenin in 15 batches of dandelion (S1–S15) were determined by ESM and QAMS, respectively. The relative conversion factors (RCF) ($f_{si}$) between the selected reference and other references in QAMS can be affected by a change in experimental conditions, such as flow rate, column temperature and standard concentration. Therefore, $f_{si}$ affect the final analysis result. The RCF ($f_{si}$) is calculated by linear regression equation, which is relatively stable (table 5). Errors caused by instruments, reagents, experimental methods or environmental conditions in the course of an experiment are relative errors (REs). RE was built between QAMS and ESM to examine the deviations using formula (3.1).

$$\mathrm{RE} = \frac{\mathrm{QAMS} - \mathrm{ESM}}{\mathrm{ESM}} \times 100\%.$$
(3.1)

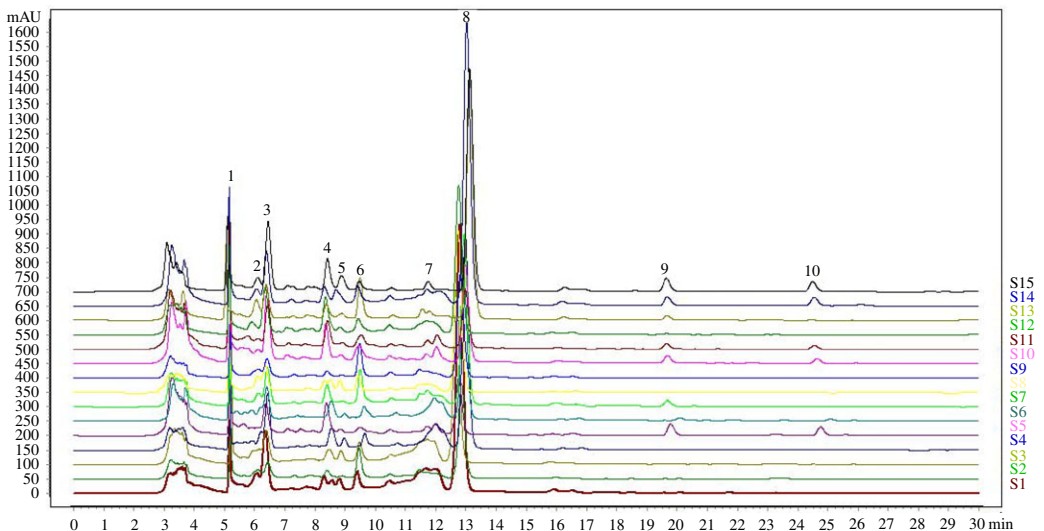

**Figure 2.** HPLC characteristic fingerprints of 15 dandelion samples. 2: chlorogenic acid, 4: caffeic acid, 7: p-coumaric acid, 8: cichoric acid, 9: luteolin and 10: apigenin.

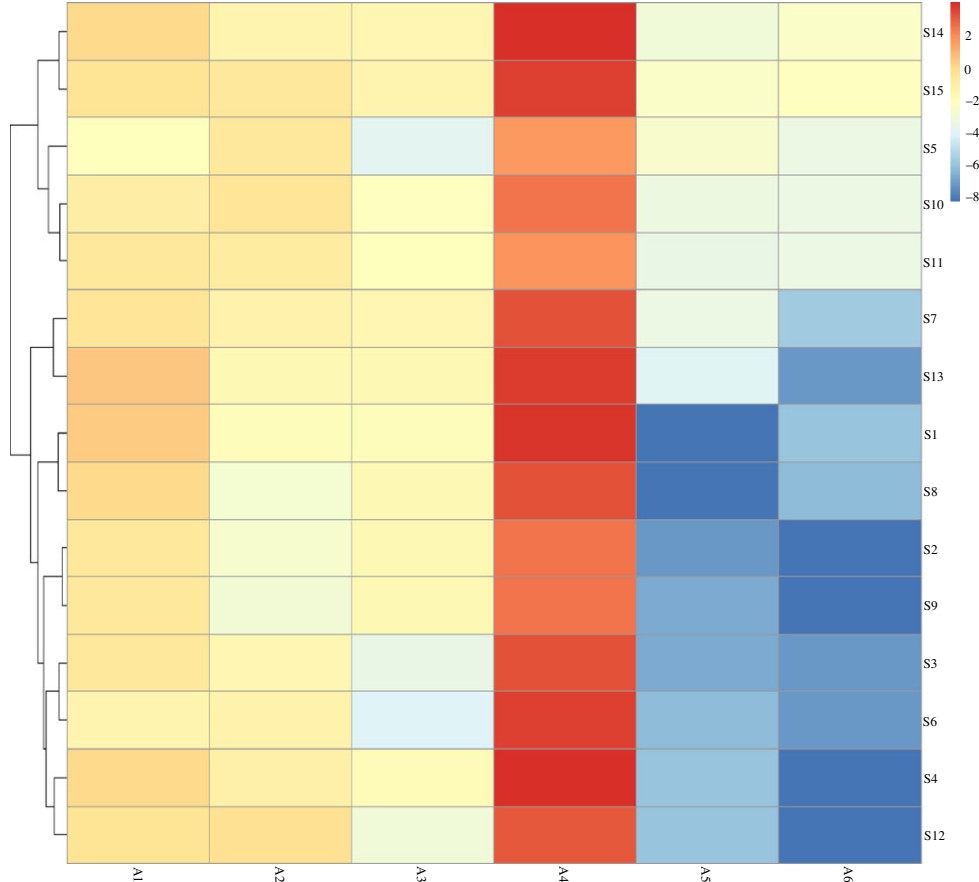

**Figure 3.** R language cluster analysis diagram of 15 dandelion samples. A1: chlorogenic acid, A2: caffeic acid, A3: p-coumaric acid, A4: cichoric acid, A5: luteolin and A6: apigenin.

The six compound contents in dandelion between two methods are shown in table 6. The changes of RE and RSDs were within the range of 5%, and there was no significant difference ($t$-test, $p > 0.05$) in quantitative determination proving the consistency of QAMS and ESM. It was observed that among these six components, the average contents of them were 0.7456, 0.4048, 0.2242, 9.1278, 0.0566 and

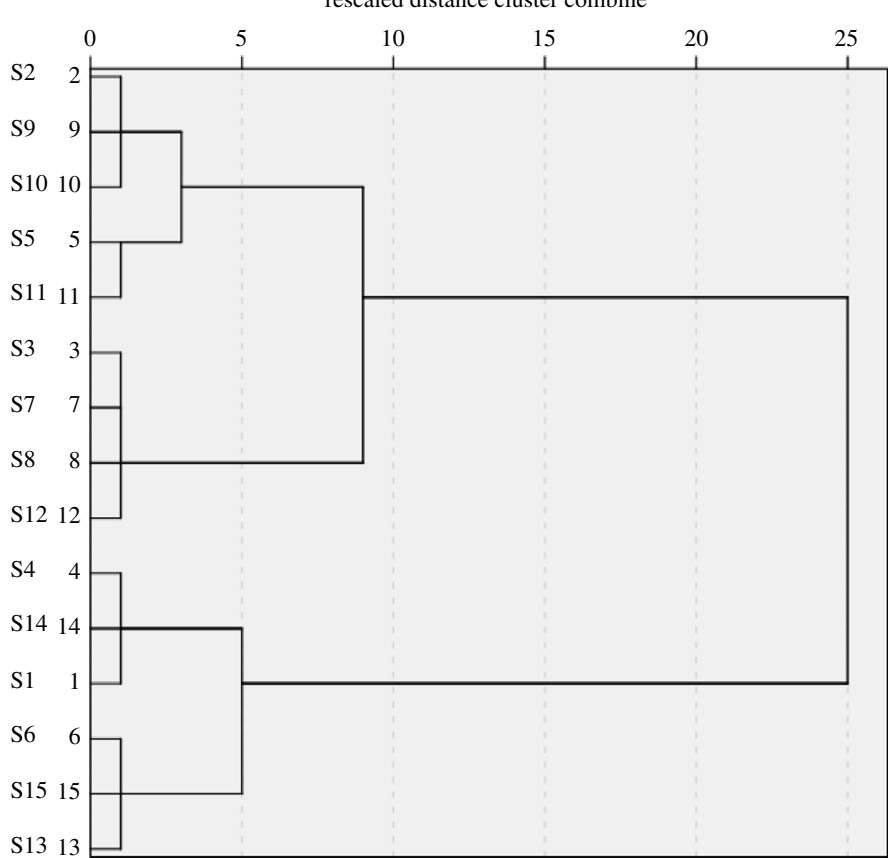

**Figure 4.** Dendrograms of HCA for the 15 tested samples of dandelion.

$0.0490 \, \text{mg g}^{-1}$, respectively. QAMS can be applied in determinating the content of a variety of components in different laboratories.

## 3.4. Quality evaluation of dandelion by fingerprint

From each of 15 batches of dandelion treatment solution was taken 10 µl for HPLC determination, and generated characteristic chromatogram of the model with 10 common peaks using the similarity evaluation system for the chromatographic fingerprint of TCM (2012) (figure 1*a*). Six common peaks (chlorogenic acid, caffeic acid, p-coumaric acid, cichoric acid, luteolin and apigenin) were identified through retention time compared with the mixed standard reference. The chromatographic fingerprint of mixed standard reference is shown in figure 1*b*. In order to obtain an eminent fingerprint, the sample (S15) of good quality is screened as the reference chromatogram. HPLC characteristic fingerprints of 15 dandelion samples are shown in figure 2. The similarity of 15 batches of dandelion samples was evaluated (table 1). As a result, their similarity values calculated were greater than or equal to 0.82, which has a high degree of fit in different regions.

## 3.5. Hierarchical cluster analysis and factor analysis result

For the sake of highlighting the differences of a dandelion from different areas, 15 batches of dandelion collected from different areas were classified by HCA according to their similarities. Moreover, R language and SPSS software were used for HCA. The results are shown in figures 3 and 4.

The R language heat map used the similarity degree of the contents of six active components in dandelion for HCA. The 15 batches of samples were mainly divided into two categories according to the similarity difference between luteolin and apigenin. S14, S15, S5, S10 and S11 were the mother category, and the rest of the batches were second category. According to the similarity difference of cichoric acid content, the first group can be also divided into two categories, S14, S15, S5 as a group and S10, S11 as a group. The R language heat map refined the content difference of

**Table 7.** Total variance explained. Extraction method: principal component analysis.

| component | initial eigenvalues | | | extraction sums of squared loadings | | | rotation sums of squared loadings | | |
|---|---|---|---|---|---|---|---|---|---|
| | total | % of variance | cumulative % | total | % of variance | cumulative % | total | % of variance | cumulative % |
| 1 | 2.717 | 45.282 | 45.282 | 2.717 | 45.282 | 45.282 | 2.280 | 38.006 | 38.006 |
| 2 | 1.836 | 30.596 | 75.878 | 1.836 | 30.596 | 75.878 | 1.663 | 27.720 | 65.727 |
| 3 | 0.804 | 13.405 | 89.283 | 0.804 | 13.405 | 89.283 | 1.413 | 23.556 | 89.283 |
| 4 | 0.414 | 6.898 | 96.181 | | | | | | |
| 5 | 0.168 | 2.803 | 98.984 | | | | | | |
| 6 | 0.061 | 1.016 | 100.000 | | | | | | |

**Table 8.** FA results of dandelion. Extraction method: principal component analysis. Rotation method: varimax with Kaiser normalization.

| | rotated component matrix[a] | | | component score coefficient matrix | | |
|---|---|---|---|---|---|---|
| standard solutions | F1 | F2 | F3 | F1 | F2 | F3 |
| chlorogenic acid | −0.176 | 0.806 | 0.428 | −0.006 | 0.453 | 0.070 |
| caffeic acid | 0.591 | −0.034 | −0.686 | 0.216 | 0.269 | −0.575 |
| p-coumaric acid | 0.192 | 0.338 | 0.869 | 0.171 | −0.062 | 0.685 |
| cichoric acid | −0.043 | 0.939 | 0.063 | 0.034 | 0.704 | −0.307 |
| luteolin | 0.977 | −0.097 | 0.004 | 0.441 | −0.026 | 0.115 |
| apigenin | 0.953 | −0.081 | −0.004 | 0.430 | −0.012 | 0.100 |

[a]Rotation converged in four iterations.

different components in different batches of dandelion. The contents of six active components in 15 selections of dandelion were taken as variables, and HCA was performed using SPSS 22.0 software, intergroup mean linking method and square Euclidean distance. When the square Euclidean distance was 5, it was divided into three groups: S2, S5, S9, S10, S11 as a group; S3, S7, S8, S12 as a group; S1, S4, S6, S13, S14, S15 as a group. The result corresponds to the FA ranking situation, and the batches with similar scores were classified into one group. From the two different HCA, it can be seen that the dandelion from the same province may not always be in the same category, which may be related to planting methods, harvesting methods, harvesting time and preliminary processing methods.

FA is to simplify the index through dimensionality reduction on the premise of keeping the original data information as much as possible. In this experiment, 10 common peak areas of 15 batches of samples were assessed by SPSS. The results are shown in table 7. The results of Kaiser–Meyer–Olkin (KMO) test and Bartlett test of sphericity show that KMO statistic is 0.542, Bartlett statistic of sphericity test is 45.487, and p-value is 0.000. It shows that the data have correlation and can be used for FA. FA was carried out after data conversion. The six factors were simplified into three main factors, and the load matrix of the rotated factors was obtained by orthogonal rotation with maximum variance. As can be seen from the table 7, the first three principal component eigenvalues are greater than 0.8, and the cumulative contribution rate of the difference is 89.283%. Therefore, multiple components of dandelion can be simplified into three principal components for analysis. The first major factor played a major role, and the contribution rate was 38.006%, which was mainly determined by luteolin and apigenin. The contribution rate of the second major factor was 27.720%, which was mainly determined by cichoric acid and chlorogenic acid. The contribution rate of the third major factor was 23.556%, which was mainly determined by p-coumaric acid and caffeic acid. According to the scoring coefficient of each factor after rotation, the scores of the first three main factors were calculated as $F1$, $F2$ and $F3$ (table 8). The comprehensive scoring model of dandelion quality, $F = (38.006F1 + 27.720F2 + 23.556F3)/89.283$, was established with the contribution rate of each major factor as the weight (table 1). The dandelion (S14) in Nanjing city, Jiangsu province, has the best quality due to the highest overall score. The overall score of dandelion in East China is higher, which is possible due to the superior natural environment conditions in this region. The terrain is mainly plain, monsoon climate and abundant water resources. Different growing environment, such as sunlight, soil and climatic conditions, havea great influence on the quality of dandelion.

## 4. Conclusion

In order to improve the quality assurance of dandelion on the basis of HPLC method, to overcome the shortage of the multi-component determination method, characteristic fingerprint combined with QAMS method was established. The similarity values of 15 selections of dandelion were calculated (greater than or equal to 0.82), which indicates that although dandelion is widely distributed, it still has a high degree of fit in different regions. The method is suitable for the determination of six active compounds in the dandelion sample. The correlation coefficient of dandelion content greater than 0.998 and RSD% less than 0.05 were determined by the single marker method and traditional ESM. HPLC-QAMS method

can get as good results as ESM. The combination of fingerprint and QAMS via MCM (HCA, FA) was a comprehensive and efficient method for quality analysis and evaluation of dandelion.

Ethics. The study was approved by the College of Chemistry, Chemical Engineering and Resource Utilization, Northeast Forestry University.

Data accessibility. The data and sources of data used in the paper are included in this submission.

The data are provided in the electronic supplementary material [32].

Authors' contributions. C.L., Y.T., C.Z., S.L., T.W., B.Q. and Y.F. substantial contributions to conception and design; Y.T. acquisition of data; Y.T. and B.Q. analysis and interpretation of data; C.L. and C.Z. drafting the article or revising it critically for important intellectual content.

Competing interests. The authors have declared no conflict of interest.

Funding. This experiment and analysis were carried out at Key Laboratory of Forest Plant Ecology, Ministry of Education, Northeast Forestry University. This work was financially supported by the Fundamental Research Fund for Central Universities (grant no. 2572019CZ01), the National Natural Science Foundation (grant no. 31870609), the 111 Project of China (B20088) and Heilongjiang Touyan Innovation Team Program (Tree Genetics and Breeding Innovation Team).

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
