## [Peer Review File · Royal Society Open Science]

Review History

RSOS-210614.R0 (Original submission)

Review form: Reviewer 1

Is the manuscript scientifically sound in its present form?

Yes

Are the interpretations and conclusions justified by the results?

Yes

Is the language acceptable?

Yes

Do you have any ethical concerns with this paper?

No

Have you any concerns about statistical analyses in this paper?

No

Recommendation?

Major revision is needed (please make suggestions in comments)

Comments to the Author(s)

The manuscript reported a quality assessment method based on Quantitative Analysis of Multi-components by Single Marker (QAMS) and fingerprint to detect the ingredients of dandelion. The novelty of the work is not clear. The data is regular and simple compared with other similar articles. On the basis of the novelty and the data, I prefer to give a major revision. Some detailed suggestions are listed:

1. The article researched the active ingredients in 15 batches of dandelion, so please supplement data to prove that there are also six active compounds in other batches of dandelion.
2. Please mark the six active compounds clearly in Figure 2.
3. Please supplement quantitative analysis data for the six active compounds and explore the quality differences.
4. There are some mistakes in Table 7 and Table 8, such as .804 , .414 and so on. In addition, the reserved digits of all data should be the same in the tables.
5. The unit in the ordinate of Figure 2 should be "mAU" and it need to indicate what the horizontal and vertical coordinates represent.
6. Please mark the legends of all figures.

Review form: Reviewer 2 (Dionisio Olmedo)

Is the manuscript scientifically sound in its present form?

Yes

Are the interpretations and conclusions justified by the results?

Yes

Is the language acceptable?

Yes

Do you have any ethical concerns with this paper?

No

Have you any concerns about statistical analyses in this paper?

No

Recommendation?

Accept with minor revision (please list in comments)

Comments to the Author(s)

Future perspective: It would be interesting to expand further work to apply this methodology to different dandelion species existing in China and are used in traditional medicine.

Review form: Reviewer 3

Is the manuscript scientifically sound in its present form?

No

Are the interpretations and conclusions justified by the results?

Yes

Is the language acceptable?

No

Do you have any ethical concerns with this paper?

No

Have you any concerns about statistical analyses in this paper?

No

Recommendation?

Major revision is needed (please make suggestions in comments)

Comments to the Author(s)

The work presented by the authors uses complex statistical tools (in this case, multivariate statistical analysis methods) to perform a joint quantitative analysis of chemical markers of the species *Taraxacum mongolicum* (phenylpropanoids and flavonoids), however, it is relevant to question whether the approach proposed by the authors is really valid or even necessary as it demands more complex and complicated data analysis. Would it not be possible to achieve the same objectives presented in this work simply by performing a quantitative analysis using internal standardization, which would allow the simultaneous quantification of the chemical markers described, in a comparable way and which demands a much simpler data analysis. performed and interpreted?

In order to demonstrate the relevance of the study carried out, there is a need for the authors to present scientific and technically plausible arguments to justify the application of the analytical approach proposed in the manuscript to the detriment of the quantification approach by internal standardization, widely used in the quality control of matters medicinal plant raw.

Decision letter (RSOS-210614.R0)

Dear Professor Zhao:

Title: Application of fingerprint combined with Quantitative Analysis and multivariate chemometric methods in quality evaluation of *Taraxacum mongolicum*
Manuscript ID: RSOS-210614

The editor assigned to your manuscript has now received comments from reviewers. We would like you to revise your paper in accordance with the referee and Subject Editor suggestions which can be found below (not including confidential reports to the Editor). Please note this decision does not guarantee eventual acceptance.

Please submit your revised paper before 28-Jul-2021. Please note that the revision deadline will expire at 00.00am on this date. If we do not hear from you within this time then it will be assumed that the paper has been withdrawn. In exceptional circumstances, extensions may be possible if agreed with the Editorial Office in advance. We do not allow multiple rounds of revision so we urge you to make every effort to fully address all of the comments at this stage. If deemed necessary by the Editors, your manuscript will be sent back to one or more of the original reviewers for assessment. If the original reviewers are not available we may invite new reviewers.

On behalf of the Subject Editor Professor Anthony Stace and the Associate Editor Dr Andrew Harned.

RSC Associate Editor:

Comments to the Author:

The work presented in this paper may prove useful to other analytical chemists working in the area of plant extracts. But, the referees have raised several valid concerns with the submitted manuscript. I ask the authors to carefully consider these concerns while crafting a revised manuscript. In particular, the authors should pay particular attention to the comments from Reviewer 3, and make a clear case for their choice of analytical methods and tools used in this paper.

RSC Subject Editor:

Comments to the Author:

(There are no comments.)

Reviewers' Comments to Author:

Reviewer: 1

Comments to the Author(s)

The manuscript reported a quality assessment method based on Quantitative Analysis of Multi-components by Single Marker (QAMS) and fingerprint to detect the ingredients of dandelion. The novelty of the work is not clear. The data is regular and simple compared with other similar articles. On the basis of the novelty and the data, I prefer to give a major revision. Some detailed suggestions are listed:

1. The article researched the active ingredients in 15 batches of dandelion, so please supplement data to prove that there are also six active compounds in other batches of dandelion.
2. Please mark the six active compounds clearly in Figure 2.
3. Please supplement quantitative analysis data for the six active compounds and explore the quality differences.
4. There are some mistakes in Table 7 and Table 8, such as .804 , .414 and so on. In addition, the reserved digits of all data should be the same in the tables.
5. The unit in the ordinate of Figure 2 should be "mAU" and it need to indicate what the horizontal and vertical coordinates represent.
6. Please mark the legends of all figures.

Reviewer: 2

Comments to the Author(s)

Future perspective: It would be interesting to expand further work to apply this methodology to different dandelion species existing in China and are used in traditional medicine.

Reviewer: 3

Comments to the Author(s)

The work presented by the authors uses complex statistical tools (in this case, multivariate statistical analysis methods) to perform a joint quantitative analysis of chemical markers of the species *Taraxacum mongolicum* (phenylpropanoids and flavonoids), however, it is relevant to question whether the approach proposed by the authors is really valid or even necessary as it demands more complex and complicated data analysis. Would it not be possible to achieve the same objectives presented in this work simply by performing a quantitative analysis using internal standardization, which would allow the simultaneous quantification of the chemical markers described, in a comparable way and which demands a much simpler data analysis. performed and interpreted?

In order to demonstrate the relevance of the study carried out, there is a need for the authors to present scientific and technically plausible arguments to justify the application of the analytical approach proposed in the manuscript to the detriment of the quantification approach by internal standardization, widely used in the quality control of matters medicinal plant raw.

Author's Response to Decision Letter for (RSOS-210614.R0)

See Appendix A.

Decision letter (RSOS-210614.R1)

Dear Professor Zhao:

Title: Application of fingerprint combined with Quantitative Analysis and multivariate chemometric methods in quality evaluation of *Taraxacum mongolicum*
Manuscript ID: RSOS-210614.R1

Thank you for submitting the above manuscript to Royal Society Open Science. On behalf of the Editors and the Royal Society of Chemistry, I am pleased to inform you that your manuscript will be accepted for publication in Royal Society Open Science subject to minor revision in accordance with the referee suggestions. Please find the reviewers' comments at the end of this email.

The reviewers and handling editors have recommended publication, but also suggest some minor revisions to your manuscript. Therefore, I invite you to respond to the comments and revise your manuscript.

Because the schedule for publication is very tight, it is a condition of publication that you submit the revised version of your manuscript before 10-Sep-2021. Please note that the revision deadline will expire at 00.00am on this date. If you do not think you will be able to meet this date please let me know immediately.

- 1) A text file of the manuscript (tex, txt, rtf, docx or doc), references, tables (including captions) and figure captions. Do not upload a PDF as your "Main Document".
- 2) A separate electronic file of each figure (EPS or print-quality PDF preferred (either format should be produced directly from original creation package), or original software format)
- 3) Included a 100 word media summary of your paper when requested at submission. Please ensure you have entered correct contact details (email, institution and telephone) in your user account
- 4) Included the raw data to support the claims made in your paper. You can either include your data as electronic supplementary material or upload to a repository and include the relevant doi within your manuscript
- 5) All supplementary materials accompanying an accepted article will be treated as in their final form. Note that the Royal Society will neither edit nor typeset supplementary material and it will

be hosted as provided. Please ensure that the supplementary material includes the paper details where possible (authors, article title, journal name).

Kind regards,
Dr Ellis Wilde
Publishing Editor, Journals

On behalf of the Subject Editor Professor Anthony Stace and the Associate Editor Dr Andrew Harned.

RSC Associate Editor

Comments to the Author:

The authors appear to have addressed most of the concerns raised by the previous reviewers. However, there still seems to be a deficiency with regard to Point 2 from Reviewer 1 ("Please mark the six active compounds clearly in Figure 2."). The authors claim these are marked, but I do not see these indicated. When I look at Figure 2, I see ten peaks marked, but there is no indication of what compounds the peaks correspond to, which is what the reviewer was asking for. Please correct this.

Reviewer comments to Author:

Author's Response to Decision Letter for (RSOS-210614.R1)

See Appendix B.

Decision letter (RSOS-210614.R2)

Dear Professor Zhao:

Title: Application of fingerprint combined with Quantitative Analysis and multivariate chemometric methods in quality evaluation of *Taraxacum mongolicum*
Manuscript ID: RSOS-210614.R2

It is a pleasure to accept your manuscript in its current form for publication in Royal Society Open Science. The chemistry content of Royal Society Open Science is published in collaboration with the Royal Society of Chemistry.

Yours sincerely,
Dr Ellis Wilde
Publishing Editor, Journals

On behalf of the Subject Editor Professor Anthony Stace and the Associate Editor Dr Andrew Harned.

RSC Associate Editor
Comments to the Author:
(There are no comments.)

Reviewer(s)' Comments to Author:

Appendix A

Response to Editor and Reviewers

Dear Editor ,

We would like to thank the editor for giving us a chance to revise the paper, and also thank the reviewers for giving us constructive suggestions which would help us both in English and in depth to improve the quality of the paper. Here our manuscript has been modified according to the editor and reviewers' suggestions.

Sincerely yours,

Chunjian Zhao, Ph.D. Professor

The following is a point-to-point response to the editor and two reviewers' comments.

RSC Associate Editor:

Comments to the Author:

The work presented in this paper may prove useful to other analytical chemists working in the area of plant extracts. But, the referees have raised several valid concerns with the submitted manuscript. I ask the authors to carefully consider these concerns while crafting a revised manuscript. In particular, the authors should pay particular attention to

the comments from Reviewer 3, and make a clear case for their choice of analytical methods and tools used in this paper.

RSC Subject Editor:

Comments to the Author:

(There are no comments.)

Reviewer #1:

1. The article researched the active ingredients in 15 batches of dandelion, so please supplement data to prove that there are also six active compounds in other batches of dandelion

>> All the 15 batches of dandelion contain six active compounds, and the supplement data have been shown in revision (see Table 6 in revised manuscript).

2. Please mark the six active compounds clearly in Figure 2.

>> The six active compounds have been clearly marked in Figure 2.

3. Please supplement quantitative analysis data for the six active compounds and explore the quality differences.

>> The quantitative analysis data for the six active compounds have been supplemented and explored the quality differences. (see Table 6 in

revised manuscript). In this experiment, 10 common peak areas of 15 batches of samples were assessed by SPSS. It shows that the data has correlation and can be used for factor analysis (FA). FA method was used to score the quality difference of 15 batches of Dandelion. The comprehensive scoring model of dandelion quality was established with the contribution rate of each major factor as the weight (see Table 1 in revised manuscript).

4. There are some mistakes in Table 7 and Table 8, such as .804 , .414 and so on. In addition, the reserved digits of all data should be the same in the tables.

The mistakes have been corrected (see Table 7 and Table 8 in revised manuscript). The reserved digits of all data have also been corrected in the tables.

5. The unit in the ordinate of Figure 2 should be “mAU” and it need to indicate what the horizontal and vertical coordinates represent.

The unit in the ordinate of Figure 2 have been revised as “mAU”. The horizontal and vertical coordinates represent have been indicated. (see Figure 2 in revised manuscript).

6. Please mark the legends of all figures.

The legends of all figures have been marked (see caption of figures in revised manuscript).

Reviewer: 2

Comments to the Author(s)

Future perspective: It would be interesting to expand further work to apply this methodology to different dandelion species existing in China and are used in traditional medicine.

Reviewer: 3

Comments to the Author(s)

The work presented by the authors uses complex statistical tools (in this case, multivariate statistical analysis methods) to perform a joint quantitative analysis of chemical markers of the species *Taraxacum mongolicum* (phenylpropanoids and flavonoids), however, it is relevant to question whether the approach proposed by the authors is really valid or even necessary as it demands more complex and complicated data analysis. Would it not be possible to achieve the same objectives presented in this work simply by performing a quantitative analysis using

internal standardization, which would allow the simultaneous quantification of the chemical markers described, in a comparable way and which demands a much simpler data analysis. performed and interpreted?

In order to demonstrate the relevance of the study carried out, there is a need for the authors to present scientific and technically plausible arguments to justify the application of the analytical approach proposed in the manuscript to the detriment of the quantification approach by internal standardization, widely used in the quality control of matters medicinal plant raw.

>> Multivariate statistical analysis method is to verify the effectiveness of the QAMS, not the necessary analysis of the QAMS, in the future the application of method will not demands so complex and complicated data analysis. Other methods can also achieve the objectives, but compared with other methods, QAMS can be used to realize the determination of multiple components of Chinese medicine by relative correction factor (RCF) in the absence of reference substances. Therefore, QAMS method is suitable for the content determination of TCM with complex pharmacodynamics. This method saves experimental consumables, simplifies operation steps and saves determination time, and the content determination results have little error.

The purpose of QAMS method is different from other measurement

methods. QAMS method integrates the advantages of existing content determination methods. Through the internal functional relationship among the components of medicinal materials, only one effective and stable component can be determined so as to realize the simultaneous determination of multiple components to be measured. It is a kind of multi-index quality evaluation mode which is suitable for the difficult preparation of reference substance, high cost or unstable reference substance. QAMS method is also widely recognized all over the world. As early as 2006, the European Pharmacopoeia contained *Ginkgo biloba* standards^[1], which adopted similar detection methods for total component analysis, as well as the determination of total astaxanthin in Japanese Pharmacopoeia^[2] and *Andrographis paniculata* in United States Pharmacopoeia^[3]. The total content of quercetin, kaempferol and isoramnetin was used to calculate the content of total flavonoid glycosides by the coefficient 2.51 under the standard item of *Ginkgo biloba* leaves in 2010 edition of Chinese Pharmacopoeia^[4].

[1]European Pharmacopoeia[S].2006, 18:436.

[2]Japanese Pharmacopoeia[S].2:24.

[3]the United States Pharmacopoeia[S]. 2008:1071.

[4] Chinese Pharmacopoeia.at one[S].2010:296, 285.

Appendix B

Response to Referees

Dear reviewer ,

We would like to thank the reviewer for giving us constructive suggestions which would help us both in English and in depth to improve the quality of the paper. Here our manuscript has been modified according to the reviewer' suggestions.

Sincerely yours,

Chunjian Zhao, Ph.D. Professor

RSC Associate Editor

Comments to the Author:

The authors appear to have addressed most of the concerns raised by the previous reviewers. However, there still seems to be a deficiency with regard to Point 2 from Reviewer 1 ("Please mark the six active compounds clearly in Figure 2."). The authors claim these are marked, but I do not see these indicated. When I look at Figure 2, I see ten peaks marked, but there is no indication of what compounds the peaks correspond to, which is what the reviewer was asking for. Please correct this.

The mistakes have been corrected. The six active compounds have been clearly marked in Figure 2 (see caption of Figure 2 in revised manuscript).

Besides, there are some mistakes in Table 3 and the regression data have been corrected (see Table 3 in revised manuscript).